# Does Asthma Disrupt Psychological Wellbeing in Pregnancy?

**DOI:** 10.3390/jcm12196335

**Published:** 2023-10-02

**Authors:** Paola C. Fernández-Paredes, Eva Morales, Concepción Lopez-Soler, Luis Garcia-Marcos

**Affiliations:** 1Department of Allergy, Arrixaca University Hospital, University of Murcia, 30120 Murcia, Spain; paola_cfp@hotmail.com; 2Department of Public Health Sciences, IMIB Bio-Health Research Institute, University of Murcia, 30120 Murcia, Spain; embarto@hotmail.com; 3Clinical Psychology Unit, Arrixaca Children’s University Hospital, University of Murcia, 30120 Murcia, Spain; clopezs@um.es; 4Paediatric Allergy and Pulmonology Units, Virgen de la Arrixaca University Children’s Hospital, IMIB Bio-Health Research Institute, University of Murcia, 30120 Murcia, Spain

**Keywords:** asthma, pregnancy, psychological test, family Apgar, Edinburgh scale, STAI inventory, NELA cohort

## Abstract

(1) Background: Asthma is a very prevalent disease with special characteristics during pregnancy, however, little is known about its relationship to the psychological wellbeing of women in this period; we aimed to know whether depression and anxiety symptoms are more frequent in asthmatic pregnant women. (2) Methods: Family Apgar (week 20), Edinburgh Postnatal Depression Scale (weeks 20 and 32) and State-Trait Anxiety Inventory (STAI) (week 32) tests were administered to 738 pregnant women (81 asthmatics) in the Nutrition in Early Life and Asthma (NELA) birth cohort. (3) Results: There were no significant differences between asthmatic and non-asthmatic pregnant women in any of the different tests at any of the time points. The mean scores for the different tests and timepoints between asthmatic and non-asthmatic pregnant women were: Apgar 20, 17.9 ± 2.2 vs. 10.0 ± 2.2; Edinburgh 20, 6.7 ± 4.2 vs. 6.9 ± 4.3; Edinburgh 32, 5.9 ± 4.4 vs. 5.6 ± 4.3; and STAI 32, 16.7 ± 8.4 vs. 15.8 ± 8.3. The proportion of pregnant women out of the normal range score for any of the tests and time points was also similar in both populations. (4) Conclusions: asthma is not associated with the psychological wellbeing of pregnant women.

## 1. Introduction

Asthma is a chronic disease characterized by chronic inflammation of the airway, in which several cells and mediators participate. It is determined by interaction between genetic and environmental factors and causes bronchial hyperresponsiveness and variable obstruction to air flow in the airways, with total or partial reversibility either spontaneously or induced with drugs [1]. It is the most prevalent respiratory disease during the first half of life, affecting about 10% of the population [2,3].

Asthma is also the most frequent respiratory disease in pregnancy with a prevalence between 8% and 13% around the world and it affects more than 10% of women of fertile ages, most of whom are sensitized to aeroallergens [4]. Many pregnant asthmatic women undergo changes in the progression of their disease: asthma worsens in about one third of them, improves in another third and is stable in the other third [5]. Asthma is the respiratory disease that most frequently complicates pregnancy and persists as a high-risk condition despite advances in treatment [6,7].

The relationship between asthma and pregnancy complications is complex due in part to asthma’s interaction with of other factors such as smoking, obesity and other comorbidities that independently impact obstetric and fetal outcomes [8].

On the other hand, mental diseases during pregnancy are thought to be as frequent as 15%, with low socioeconomic status as a risk factor [9]. Anxiety and depression are common conditions that affect a considerable part of the population in many countries [10]. Several studies show that, among pregnant women, depression ranges from 6% to 13% [11,12,13]. Both anxiety and depression are common not only during pregnancy but also post-partum and their symptoms can be mild or severe [14].

Mental diseases have been also recognized as common comorbidities in asthmatic patients, with anxiety, depression and panic disorders more frequent among asthmatic individuals than among the general population [9].

To the best of our knowledge, the impact of mental diseases in asthma control during pregnancy has been previously reported only by one study in an Australian cohort of pregnant women. This study suggests that maternal anxiety is associated with poorer asthma control (including an increased number of exacerbations) and lower quality of life [15]. However, no study has yet compared the psychological wellbeing of asthmatic and non-asthmatic pregnant women.

The aim of the present study is to know whether depression and anxiety as measured by the Family Apgar, Edinburgh and STAI tests are more frequent in asthmatic pregnant women than non-asthmatic ones.

## 2. Materials and Methods

The group of pregnant women in the present study are participants in the “Nutrition in Early Life and Asthma” (NELA) birth cohort. The details of this cohort have been reported in detail previously [16]. Briefly, the NELA study recruited pregnant women at the 20th week of gestation when they attended to the “Virgen de la Arrixaca” University Clinical Hospital (Murcia, Spain) for their routine echography control. They were followed up during pregnancy and beyond. During gestational follow-up a section on psychological wellbeing, including the Family Apgar, Edinburgh Postnatal Depression Scale and/or State-Trait Anxiety Inventory (STAI), depending on the visit, was included at weeks 20 and 32.

Asthma in pregnant women was defined as a positive response to the question “Have you ever been diagnosed of asthma” which was asked in the questionnaire at the 20-week recruitment visit.

The family Apgar test, designed to assess family function by primary health care teams, explores satisfaction in family relationships [17] and has good validity and reliability [18]. It examines five basic functions of the family, which are: adaptability, partnership, growth, affection, and resolve. Scores from this test are highly correlated to anxiety and depression [19] and follow a Likert scale either from 0 to 2 (10 points maximum) or from 0 to 4 (20 points maximum). With a maximum score of 20, 0 to 16 indicates dysfunctionality (from mild, 14–17; to moderate, 10–13; and severe, ≤9) and 17 to 20 functionality [20,21].

The Edinburgh Postnatal Depression Scale has been validated in many settings and populations [22], and also for its use during pregnancy [23]. It measures depression and scores from 0 to 30 over 10 items which rate from 0 to 3 also in a Likert scale. Scores from 10 to 30 indicate depression.

The State-Trait Anxiety Inventory (STAI) has been also validated in diverse settings and populations and specifically perinatally [24]; it assesses anxiety in two dimensions: state and trait, although it also detects depression [24]. In the present study, STAI was used to assess traits. Its score (trait, 20 items) ranges from 0 to 60 and for adult women and its 50th percentile is 22–23.

In Spanish-speaking populations, Cronbach’s alpha reliability has been found to be 0.90 for trait and 0.94 for state anxiety with the STAI test [25]; 0.82 for the family Apgar test [26]; and 0.78 for the Edinburgh test [27].

Apart from the results of the three psychological tests at the different time points, other variables available in the study which were included in the analyses as covariates were age; living area (urban, residential, countryside); civil state (married or stable partner, single, divorced or separated); education level (basic, secondary, high, university); social class (as per job classification in Domingo-Salvany et al. [28]) grouped in I–II, III, IV–V and no current job; number of previous pregnancies and abortions; current smoking (yes or no); and use of inhaled corticosteroids in 1st, 2nd or 3rd trimester (yes or no).

The results of the psychological tests were compared between asthmatic and non-asthmatic mothers by means of a t-test (for the scores taken as continuous variables) and using the chi-squared test when tests scores were categorized into “inside the normal range” vs. “out of the normal range”, with 0 = inside and 1 = outside. This analysis was not conducted for the STAI test as the scores are continuous and a normal range is not established.

To detect a significant a coefficient of determination (R^2^) in the different multivariate models of at least 0.1, testing the 12 variables to be included in the models for the three psychological tests with a probability level of 0.05 and a power level of 0.8, the needed sample size was 167 subjects.

To control for variables which could modify the association of asthma with the psychological test results, multiple linear regression analyses or logistic regression analyses were performed using test scores as dependent variables (either as continuous or as dichotomous variables), and asthma as the independent variable, and those variables already defined as covariates. As the purpose of the regressions was not to predict the result of the psychological tests but rather assess the size of the effect (if any) of asthma in pregnant women on the scores of the different psychological tests (taken either numerically or categorized in normal/not normal), allowing for other variables with potential influence on the outcome and on the risk factor (asthma), goodness of fit was not taken into account.

All statistical calculations were performed using STATA/SE v18 software (College Station, TX, USA).

All procedures included in the NELA cohort study were approved by the ethics committee of the “Virgen de la Arrixaca” University Clinical Hospital (report 9/14; 29 September 2014).

## 3. Results

### 3.1. Study Population

A total of 1350 women were invited to participate in the NELA study and 738 (54%) were ultimately followed up at least until delivery and included in the present study. This group of 738 individuals was the total sample size; however, depending on the missing values of certain variables in certain individuals, multiple regression analyses were carried out with slightly lower numbers, which ranged from 626 to 671. These numbers would be able to display any significant (*p* < 0.001) association of any dependent variable with the outcome with a power level of 0.99 for the lowest coefficient of determination found (R^2^ = 0.085 in Edinburgh test at 20 week of gestation). Of the total sample size of 738, 81 (11.0%) had asthma as diagnosed by a doctor when they were recruited. 

The demographics of asthmatic and non-asthmatic pregnant women are shown in Table 1. Both groups were comparable in all factors such as age, living area, civil state, education years, social class, previous abortions or pregnancies and smoking habit. As expected, the use of inhaled corticosteroids was significantly more frequent among asthmatic mothers.

### 3.2. Differences in Psychological Tests 

There were no statistically significant differences in any of the psychological scores for any of the tests performed at the different time points, either when scores were taken as continuous variables nor when they were categorized into “in the normal range” versus “out of the normal range”. In fact, quantitative scores were very similar between both groups of pregnant women, with the largest difference found (statistically non-significant) in the STAI test at week 32 (16.7 ± 8.4 in asthmatics vs. 15.8 ± 8.3 in non-asthmatics, *p* = 0.378) (Table 2 and Table 3).

### 3.3. Multivariate Analyses

Table 4 and Table 5 show the factors that had an association with the scores (either as continuous or as dichotomous variables) in the simple or multiple regression or logistic regression analyses. It is clearly shown that asthma is not associated either with the numerical results of the tests or with the qualitative ones (in or out of the normal range). This lack of association occurs both with the univariable and the multivariable analyses. The coefficients of the two Edinburgh tests (weeks 20 and 32) and the STAI scale tend to be related to the civil state of “divorced/separated” (Table 4). However, when the cut-off point of “in or out of the normal range” is applied and the logistic regression performed, these associations disappear, most probably due to the number of divorced or single pregnant women included in the analyses being very few (8 single, 2 divorced). It should be clarified that in contrast to the previous analysis (Table 2 and Table 3) multivariate analyses were not stratified by asthma but considered all the individuals as a population sample.

Having completed university studies (as compared to no or basic educations) was associated with higher family Apgar scores when the result of the test was considered as a numeric continuous variable. However, when the cut-off point was established, the bivariable analysis remained significant but the multiple did not. This is also the case for the Edinburgh test. Note that for the logistic regression analyses, and as said previously, the outcome variable “inside/outside the normal range” was taken as 0/1. Not having any current job was associated with worse scores for outside the normal ranges but mainly in the bivariable analyses. Only a weak association was maintained in the multiple analysis of the logistic regression for the Edinburgh test in week 32 (aOR: 2.03; CI95%: 1.02;4.01). Being a current smoker was significantly associated with worse scores in all tests both in the bivariable and multiple linear and logistic regressions. Taking inhaled corticosteroids in the third trimester of pregnancy was associated with some tests results (Table 2 and Table 3).

## 4. Discussion

In the present study of depression and anxiety in asthmatic pregnant woman compared to their non-asthmatic peers, the results show that those psychological conditions are equally frequent in both populations. To the best of our knowledge, the information about the relationship between psychological and allergic diseases in pregnant women is very limited. Most of the literature found in this area focuses on this association but in the general population, not specifically in pregnancy, showing that asthma, especially when severe, is associated with depression 38.

Factors that, in the present study, have significantly affected the psychological tests results are mainly socio-demographic ones, such as study level, social class and civil state. However, it should be underlined that there was no individual divorced or separated in the group of asthmatic pregnant women, rendering any result in the category of civil state meaningless. Compared to having no or basic educations, pregnant women with university studies tended to have functioning families in the bivariate analyses, although the power of the association dropped when the results were adjusted (aOR;95%CI for unfunctional family in this category: 0.65, 0.28;1.50). The same trend was found for the results of the Edinburgh tests at 20 and 32 weeks of gestation (aOR;95%CI for depression/anxiety in this category at week 20: 0.46, 0.21;1.01 and at week 32: 0.72 0.31;1.69). Following the same trend, women of lower social classes tended to belong to dysfunctional families or to more likely have depression or anxiety compared to higher classes. As previously seen, the power of the association dropped when the variables were adjusted. In the general population, high job control (more frequent among higher rank jobs included in social classes I and II) reduces the risk of developing depression even when correcting for socio-demographic factors and previous psychiatric diseases [29,30]. Similarly, university studies are a protective factor for depression and anxiety and this accumulates throughout life [31]. Thus, in this regard, pregnant women behave similarly to the general population and asthma does not seem to be a factor acting differently here to drive to family dysfunctionality and personal depression or anxiety than in the general population.

A factor that does not seem to be related to higher rates of depression or anxiety and family dysfunctionality, in our population of pregnant mothers, is previous abortions. Previous abortions have been shown to be a factor predisposing women to depression in the following pregnancy in some populations [32,33,34]. To what extent asthma, which is not considered in the aforementioned studies, could have modified their results is difficult to say, but under the light of the results of the present study it does not seem that consideration of that condition would have greatly modified the results. Why previous abortions in the women in our study are not associated with test scores pointing to depression or anxiety and family dysfunctionality is probably explained by the differences in the sample population, in the tests used and in the relatively low number of women with previous abortions. Additionally, the number of abortions in asthmatic and non-asthmatic women was similar and, for the aims of this study, abortions did not modify the effect (if any) of asthma on the different psychological tests.

Older age and smoking habits were somewhat associated with scores indicating depression/anxiety and family dysfunctionality. For instance, for every additional year, the probability of a pregnant women having a score indicating family dysfunctionality increased about 5% (aOR 1.05, 1.00;1.11). Both the Edinburgh test at 20 and 32 weeks and the STAI tests had similar results, which were not overtly significant from the statistics point of view but were quite close to it and showing a clear trend. Being a current smoker was clearly and consistently associated with scores out of the range of normality, doubling the probabilities of a score indicating depression or anxiety and family dysfunctionality. Smoking and older age have been shown to be associated with depression. As found by a systematic review which included 45 previous studies, depression was associated with smoking in adults, adolescents and even children 10–12 years old [35]. Older age in our population probably means one or several previous children which might be combined with other accompanying circumstances such as economic constraints that can make pregnant women more susceptible to depression [36]. 

The profile developed here has pregnant mothers being more prone to non-normal test scores when older, low social class, not having university studies and smokers, and looks quite like the one shown in the only study of the influence of asthma on the quality of life of pregnant mothers. The Preventing Atopy Dermatitis and Allergy in Children (PreventADALL) recruited 2697 pregnant women at the 18th week of gestation. Among the battery of questionnaires and tests, the researcher included the perceived stress scale (PSS) which was administered between the 18th and 24th weeks of gestation which was filled out by 2164 pregnant women. At mid-pregnancy, the factors significantly different between those with low (<29) and high (≥29) PSS were high income (higher proportion in lower PSS); having been previously pregnant (higher proportion in higher PSS); and cigarette smoking (higher proportion in higher PSS). Asthmatics, although having higher proportion with a PSS ≥ 29, were not statistically significant. Asthma and other allergic diseases did contribute to significant stress when they were severely symptomatic. We did not have the information about the degree of asthma control for the pregnant women in our population, and the use of inhaled corticosteroids did not show any consistent result in any of the three trimesters of pregnancy. Although using different tests to measure stress, the results of Olsson et al. from the PreventADALL cohort show a similar picture to that of the NELA cohort.

One possible limitation of the study could be the multiple comparisons scenario which might incline the results to some type I error situations. However, we do not think this is an important issue in this case as: (1) the differences between the asthmatic and non-asthmatic groups in the bivariate analyses are almost inexistent, apart from the use of inhaled corticosteroids in the asthmatics, an obvious one; (2) in multiple regression analyses multiple testing is part of the procedure to know whether there is any independent variable which can independently explain any significant variance in the dependent one; (3) the number of tests performed in those analyses (in order to make a correction such as Bonferroni’s) is difficult to know; (4) *p*-values are not of importance, but rather the sizes of the effects (coefficients or adjusted odds ratios) of the different independent, and adjusted for each other, variables in the model are; and (5) the results of the linear and logistic regressions are quite consistent, probably indicating that the significant associations found in the adjusted coefficients or odds ratios are not a result of the multiple comparisons.

There is still considerable room for specifically designed studies to address this important question regarding how asthma and other allergic diseases and their control could affect the psychological wellbeing of pregnant women and their effects on their children.

## 5. Conclusions

Using three different psychological tests at two time points in pregnancy, in a cohort of women from the general population, it does not seem that asthma is associated with depression or anxiety or family dysfunctionality. In fact, the associations found in the present study, of tests scores outside of the range of socio-demographic factors, are quite coincidental with previous studies in the general population.

## Figures and Tables

**Table 1 jcm-12-06335-t001:** Socio-demographic characteristics of the studied populations: differences between asthmatic and non-asthmatic pregnant women (*n* and percentage in brackets unless indicated).

	Asthmatics *n* = 81	Non-Asthmatics *n* = 657	*p* *
Age (mean ± SD)	32.7 ± 4.6	32.6 ± 4.7	0.859
Living area			
Urban	55 (67.9)	479 (72.9)	
Peri-urban	13 (16.0)	94 (14.3)	
Countryside	13 (16.0)	84 (12.8)	0.611
Civil state			
Married or stable partner	81 (100)	647 (98.5)	
Single	0	8 (1.2)	
Divorced/separated	0	2 (0.3)	0.535
Education			
None, basic or primary (8 years or less)	4 (4.9)	53 (8)	
Middle or incomplete secondary (9–11 years)	11 (13.6)	78 (11.9)	
Complete middle or secondary and high (12+ years)	19 (23.5)	172 (26.2)	
University	47 (58.0)	354 (53.9)	0.677
Social class			
I-II	31 (38.3)	233 (35.5)	
III	19 (23.5)	148 (22.5)	
IV-V	18 (22.2)	126 (19.2)	
No current job	13 (16.0)	150 (22.8)	0.566
Previous abortions [median (IQR)]	0 (0–1)	1 (0–1)	0.863
Previous pregnancies [median (IQR)]	1 (0–1)	1 (0–2)	0.325
Current smoker	15 (18.5)	113 (17.2)	0.767
Inhaled corticosteroids 1st trimester	6 (8.2)	2 (0.3)	<0.001
Inhaled corticosteroids 2nd trimester	6 (8.2)	6 (1)	<0.001
Inhaled corticosteroids 3rd trimester	7 (9.6)	7 (1.2)	<0.001

SD: standard deviation; IQR: interquartile range. * *p* value derived from t-test for continuous variables and from Chi2 test for categorical variables.

**Table 2 jcm-12-06335-t002:** Psychological tests * results in asthmatic and non-asthmatic pregnant women. Scores taken as continuous variables.

	Nº Individuals Tested	Asthmatics *n* = 81	Non-Asthmatics *n* = 657	*p*
Family Apgar (week 20)	733	17.9 ± 2.2	18 ± 2.2	0.868
Edinburgh test (week 20)	723	6.7 ± 4.2	6.9 ± 4.3	0.800
Edinburgh test (week 32)	655	5.9 ± 4.4	5.6 ± 4.3	0.634
STAI test (week 32)	626	16.7 ± 8.4	15.8 ± 8.3	0.378

* See text for tests explanation.

**Table 3 jcm-12-06335-t003:** Psychological tests * results in asthmatic and non-asthmatic pregnant women. Scores taken as in or out of the normal range.

	Asthmatics *n* = 81	Non-Asthmatics *n* = 657	*p*
Family Apgar (week 20)			
Functional family ^1^	65 (81.2)	529 (81)	0.959
Dysfunctional family ^1^	15 (18.7)	124 (18.9)	0.959
Edinburgh test (week 20)			
No depression ^2^	61 (78.2)	490 (76.0)	0.661
Depression ^2^	17 (21.8)	155 (24.0)	0.661
Edinburgh test (week 32)			
No depression ^2^	56 (81.2)	484 (82.6)	0.767
Depression ^2^	13 (18.8)	102 (17.4)	0.767

* See text for explanation of tests. ^1^ Functional family 17/20; dysfunctional family 0/16. ^2^ No depression 0/9; depression 10/30. Note that not every row adds up to 738 due to missing values.

**Table 4 jcm-12-06335-t004:** Associations of asthma and other factors with numerical results of the psychological tests in the simple (Coeff and 95%CI) and multiple linear regression (aCoeff. and 95%CI) analyses (all variables in the table included in the multiple analysis).

	Family Apgar		Edinburgh 20		Edinburgh 32		STAI	
	Coef	aCoef	Coef	aCoef	Coef	aCoef	Coef	aCoef
Asthma	−0.04	0.13	−0.13	−0.17	0.26	0.03	0.96	0.60
	−0.55; 0.46	−0.37; 0.63	−1.15; 0.89	−1.24; 0.90	−0.8; 1.34	−1.06; 1.12	−1.18; 3.09	−1.51; 2.70
Living area								
Urban	1	1	1	1	1	1	1	1
Residential	0.09	0.07	−0.52	−0.08	−0.04	0.18	−1.96	−1.34
	−0.36; 0.55	−0.37; 0.51	−1.43; 0.38	−1.01; 0.85	−0.99; 0.91	−0.76; 1.12	−3.83; 0.10	−3.15; 0.47
Countryside	−0.18	−0.09	0.44	0.47	0.72	0.57	1.78	0.93
	−0.65; 0.29	−0.54; 0.36	−0.50; 1.38	−0.47; 1.41	−0.25; 1.69	−0.39; 1.52	−0.15; 3.72	−0.94; 2.81
Civil state								
Married or stable partner	1	1	1	1	1	1	1	1
Single	0.80	0.84	2.20	1.39	2.36	1.47	4.93	3.20
	−0.71; 2.31	−0.64; 2.33	−0.80; 5.20	−1.72; 4.51	−0.84; 5.57	−1.68; 4.62	−1.21; 11.08	−2.77; 9.16
Divorced/separated	1.55	1.93	**7.70**	**6.94**	**6.36**	5.68	**25.22**	**23.21**
	−1.47; 4.57	−0.84; 4.71	**1.72 13.68**	**1.14; 12.7**	**0.39; 12.34**	−0.19; 11.5	**13.8; 36.7**	**12.1; 34.3**
Age	−0.01	−0.05	−0.10	0.01	−0.08	0.02	−0.22	−0.00
	−0.04; 0.03	−0.09; 0.01	−0.17; 0.03	−0.07; 0.09	−0.15; 0.00	−0.06; 0.11	−0.37; 0.07	−0.17; 0.16
Education								
None, basic or primary (8 years or less)	1	1	1	1	1	1	1	1
Middle or incomplete secondary (9–11 years)	−0.30	−0.42	0.86	0.61	1.17	1.09	1.94	1.73
	−1.01; 0.41	−1.15; 0.30	−0.59; 2.30	−0.94; 2.15	−0.35; 2.69	−0.45; 2.63	−1.09; 4.97	−1.27; 4.75
Complete middle or secondary and high (12 + y)	0.45	0.18	−0.02	0.35	0.24	0.52	0.41	1.10
	−0.19; 1.08	−0.47; 0.84	−1.30; 1.26	−1.04; 1.74	−1.08; 1.56	−0.87; 1.91	−2.27; 3.09	−1.66; 3.85
University	**1.03**	**0.69**	−1.76	−1.13	−1.30	−0.58	−3.30	−1.14
	**0.43; 1.62**	**0.02; 1.36**	−2.96; 0.55	−2.56; 0.31	−2.53; 0.07	−2.02; 0.85	−5.79; 0.80	−3.95; 1.67
Social class								
I–II	1	1	1	1	1	1	1	1
III	−0.38	−0.11	0.43	−0.46	0.66	0.00	2.41	1.01
	−0.79; 0.41	−0.55; 0.32	−0.40; 1.25	−1.37; 0.45	−0.20; 2.34	−0.93; 0.94	0.72; 4.11	−0.79; 2.81
IV–V	**−0.87**	−0.38	**1.33**	−0.14	**1.41**	0.32	**3.48**	1.21
	**−1.31; −0.43**	−0.89; 0.13	**0.45; 2.20**	−1.23; 0.94	**0.49; 2.34**	−0.78; 1.43	**1.70; 5.26**	−0.89; 3.33
No current job	**−0.51**	0.14	**1.87**	0.41	**1.62**	0.70	**4.48**	**2.21**
	**−0.93; −0.08**	−0.35; 0.64	**1.03; 2.72**	−0.63; 1.45	**0.72; 2.53**	−0.36; 1.76	**2.68; 6.28**	**0.18; 4.25**
Previous abortions	−0.14	0.25	−0.03	−0.08	−0.19	−0.02	−0.26	−0.26
	−0.36; 0.07	−0.08; 0.58	−0.46; 0.39	−0.78; 0.62	−0.64; 0.27	−0.72; 0.68	−1.16; 0.63	−1.64; 1.11
Previous pregnancies	**−0.19**	−0.23	0.03	−0.06	**−0.14**	−0.25	−0.23	−0.19
	**−0.3; −0.05**	−0.46; −0.00	−0.24; 0.30	−0.54; 0.43	**−0.43; 0.15**	−0.74; 0.24	−1.19; 0.73	−1.16; 0.77
Current smoker	**−1.09**	**−0.94**	**1.98**	**1.71**	**2.30**	**1.82**	**4.72**	**3.48**
	**−1.50; −0.68**	**−1.36; −0.51**	**1.16; 2.81**	**0.81; 2.62**	**1.43; 3.18**	**0.91; 2.73**	**3.00; 6.45**	**1.72; 5.24**
Inhaled corticosteroids 1st trimester	**−1.64**	−0.70	1.00	0.42	1.34	−0.40	5.02	3.20
	**−3.10; −0.19**	−2.29; 0.89	−1.97; 3.98	−2.92; 3.77	−1.68; 4.35	−3.76; 2.97	−0.80; 10.85	−3.20; 9.60
Inhaled corticosteroids 2nd trimester	−1.10	0.07	1.20	0.63	1.60	0.76	1.19	−1.17
	−2.30; 0.09	−1.21; 1.36	−1.34; 3.75	−2.23; 3.49	−0.87; 4.07	−1.97; 3.49	−3.59; 5.97	−6.36; 4.01
Inhaled corticosteroids 3rd trimester	**−2.55**	**−2.2**	1.18	0.38	**2.88**	2.26	4.48	2.69
	**−3.65; −1.46**	**−3.38; −1.02**	−1.08; 3.43	−2.11; 2.88	**0.60; 5.16**	−0.25; 4.76	−0.10; 9.07	−2.27; 7.65

**Table 5 jcm-12-06335-t005:** Associations of asthma and other factors with functional normal score ranges in the psychological tests in the simple (OR and 95% CI) and multiple logistic regression (aOR and 95%CI) analyses (all variables in the table included in the multiple analysis).

	Family Apgar		Edinburgh Week 20		Edinburgh Week 32	
	OR	aOR	OR	aOR	OR	aOR
Asthma	0.98	0.89	0.88	0.82	1.10	1.14
	0.54; 1.78	0.44; 1.82	0.50; 1.55	0.42; 1.62	0.58; 2.09	0.58; 2.26
Living area						
Urban	1	1	1	1	1	1
Residential	0.86	0.88	0.74	0.82	0.91	0.98
	0.50; 1.48	0.47; 1.64	0.44; 1.25	0.45; 1.50	0.50; 1.66	0.52; 1.85
Countryside	0.61	0.55	1.17	1.13	1.27	1.20
	0.33; 1.14	0.28; 1.08	0.71; 1.91	0.66; 1.94	0.72; 2.22	0.67; 2.14
Civil state						
Married or stable partner	1	1	1	1	1	1
Single	0.60	0.70	1.95	1.66	3.61	2.80
	0.07; 4.96	0.08; 6.29	0.46; 8.23	0.34; 8.13	0.80; 16.38	0.57; 13.4
Divorced/separated	NA	NA	3.24	2.91	4.82	4.69
			0.20; 52.1	0.17; 48.9	0.30; 77.6	0.27; 81.6
Age	1.01	1.05	0.99	1.04	0.97	1.01
	0.97; 1.05	1.00; 1.11	0.95; 1.02	0.99; 1.09	0.93; 1.01	0.96; 1.07
Education						
None, basic or primary (8 years or less)	1	1	1	1	1	1
Middle or incomplete secondary (9–11 years)	1.25	1.76	0.95	0.83	1.04	1.01
	0.60; 2.60	0.76; 4.14	0.46; 1.97	0.36; 1.92	0.45; 2.37	0.42; 2.41
Complete middle or secondary and high (12 + y)	0.62	0.87	0.89	1.07	0.82	0.90
	0.32; 1.23	0.39; 1.95	0.46; 1.70	0.51; 2.24	0.39; 1.70	0.40; 2.00
University	**0.41**	0.65	**0.39**	0.46	**0.46**	0.72
	**0.21; 0.77**	0.28; 1.50	**0.21; 0.74**	0.21; 1.01	**0.23; 0.92**	0.31; 1.69
Social class						
I–II	1	1	1	1	1	1
III	1.45	1.23	1.33	0.82	1.78	1.61
	0.86; 2.46	0.66; 2.30	0.82; 2.16	0.46; 1.48	1.02; 3.15	0.86; 3.01
IV–V	**2.19**	1.59	**1.76**	0.90	**1.95**	1.49
	**1.31; 3.66**	0.79; 3.20	**1.08; 2.87**	0.47; 1.76	**1.08; 3.51**	0.72; 3.08
No current job	**1.79**	1.00	**2.27**	1.36	**2.54**	**2.03**
	**1.07; 3.00**	0.49; 2.02	**1.43; 3.60**	0.73; 2.10	**1.46; 4.44**	**1.02; 4.01**
Previous abortions	1.15	0.80	1.06	1.00	0.93	0.99
	0.91; 1.45	0.52; 1.21	0.85; 1.33	0.67; 1.50	0.70; 1.25	0.63; 1.58
Previous pregnancies	1.21	1.27	1.10	0.97	0.96	0.87
	1.05; 1.40	0.94; 1.71	0.96; 1.27	0.73; 1.30	0.80; 1.15	0.63; 1.20
Current smoker	**2.76**	**2.54**	**2.25**	**2.14**	**2.39**	**2.03**
	**1.78; 4.23**	**1.54; 4.20**	**1.48; 3.40**	**1.32; 3.47**	**1.49; 3.84**	**1.22; 3.38**
Inhaled corticosteroids 1st trimester	1.49	0.59	1.18	1.29	0.67	0.11
	0.30; 7.49	0.04; 7.76	0.23; 5.91	0.18; 9.33	0.08; 5.48	0.01; 2.04
Inhaled corticosteroids 2nd trimester	1.48	0.67	1.33	1.32	2.40	4.43
	0.40; 5.61	0.10; 4.61	0.35; 5.09	0.26; 6.69	0.71; 8.10	0.88; 18.0
Inhaled corticosteroids 3rd trimester	**6.28**	**8.34**	0.96	**0.70**	1.91	1.84
	**2.14; 18.46**	**2.08; 33.5**	0.26; 3.50	**0.15; 3.30**	0.59; 6.20	0.44; 7.75

NA: not available due to very low number of individuals.

## Data Availability

The study protocol including all questionnaires and measurements of all visits already performed are available in the NELA website (https://nela.imib.es/, accessed on 2 July 2023 ) upon request to the corresponding author. The NELA data, including de-identified individual participant data, will be made available to interested researchers by the NELA Steering Committee. Access will require a formal request, a written proposal, and a signed data access agreement.

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
