# Peer review of "Does Asthma Disrupt Psychological Wellbeing in Pregnancy?"

_jcm, 2023, doi:10.3390/jcm12196335_

Round 1

Reviewer 1 Report

Methods: The authors have not provided information on the application of Cronbach's alpha for the scales utilized.

  1. The authors didn't mention the total sample size used and whether a power calculation was performed to ensure adequate statistical power should be stated.

  2.  

  3. Furthermore, briefly mention assumptions for chosen tests (t-test, Chi-squared) and explain how these were met or addressed. It would be nice if authors state whether any correction for multiple comparisons was applied.

  4. Report effect sizes (e.g., Cohen's d) to provide practical significance.

  5. Please describe how model fit was assessed and provide a concise interpretation of model coefficients.

  6. Please specify the statistical software used for analysis, including version.

  7. If applicable, mention ethical approvals or considerations related to the analysis.

The write-up demonstrates average quality in English and offers room for improvement.

Reviewer 2 Report

An interesting study evaluating the psychological effects of asthma in pregnancy. It is a Cohort study, thus women with asthma are less than women without asthma.

The authors used three scales to evaluate anxiety a) Family Apgar, b) Edinburgh Postnatal Depression Scale and c). State-Trait Anxiety Inventory (STAI). They found that there was no significant differences between asthmatic and non-asthmatic pregnant women in any of the different tests at any of the time points

A few comments:

1.       in the abstract please spell all abbreviations such as STAI

2.       In the statistical analysis methodology, it is applied the t-test, however it is not mentioned if any normality test was applied. The appropriate method before applying t-test is to ensure normality of the variables in both compared groups. Thus is essential to ensure data normality or apply an non-parametric test such as Mann Whitney U test.

3.       There are few typos, for example continuus, please revise the manuscript with the help of a language correction tool

4.       In table 2 and in general in the study it is not clear if both tests (i.e. Family Apgar and Edinburgh test) were applied in week 20 (or in 32 for Edinburgh and STAI). Propose to add a new column with the number of available test per row despite the number of cases is presented in the column headings.

5.       Tables 4 and 5 showing simple or multiple regression outcomes are hard to read please add an asterisk or use bold font for the coefficients or ORs that there is statisrical significance.

Round 2

Reviewer 1 Report

I'd like to appreciate the thorough responses provided. In regard to comment 3 concerning the multiple comparisons, I share the same perspective as the authors. Given that it may not be a critical concern, it is ok to omit or shorten the detailed explanation. I am leaving the decision on whether to retain those lines to the authors' discretion. I don't have any additional comments or suggestions. Wishing you the best of luck!"